# Structural and Molecular Changes of Human Chondrocytes Exposed to the Rotating Wall Vessel Bioreactor

**DOI:** 10.3390/biom14010025

**Published:** 2023-12-24

**Authors:** Paul Steinwerth, Jessica Bertrand, Viviann Sandt, Shannon Marchal, Jayashree Sahana, Miriam Bollmann, Herbert Schulz, Sascha Kopp, Daniela Grimm, Markus Wehland

**Affiliations:** 1Department of Microgravity and Translational Regenerative Medicine, University Clinic for Plastic, Aesthetic and Hand Surgery, Otto von Guericke University, 39106 Magdeburg, Germany; paul.steinwerth@st.ovgu.de (P.S.); viviann.sandt@med.ovgu.de (V.S.); shannon.marchal@med.ovgu.de (S.M.); herbert.schulz@med.ovgu.de (H.S.); markus.wehland@med.ovgu.de (M.W.); 2Department of Orthopaedic Surgery, Otto-von-Guericke-University Magdeburg, 39120 Magdeburg, Germany; jessica.bertrand@med.ovgu.de (J.B.); miriam.bollmann@gu.se (M.B.); 3Research Group “Magdeburger Arbeitsgemeinschaft für Forschung unter Raumfahrt-und Schwerelosigkeitsbedingungen” (MARS), Otto von Guericke University, 39106 Magdeburg, Germany; sascha.kopp@ovgu.de; 4Department of Biomedicine, Aarhus University, Ole Worms Allé 4, 8000 Aarhus, Denmark; jaysaha@biomed.au.dk; 5Core Facility Tissue Engineering, Otto von Guericke University, 39106 Magdeburg, Germany

**Keywords:** cartilage, tissue engineering, rotating wall vessel, simulated microgravity, scaffold-free, extracellular matrix, gene expression

## Abstract

Over the last 30 years, the prevalence of osteoarthritis (OA), a disease characterized by a loss of articular cartilage, has more than doubled worldwide. Patients suffer from pain and progressive loss of joint function. Cartilage is an avascular tissue mostly consisting of extracellular matrix with embedded chondrocytes. As such, it does not regenerate naturally, which makes an early onset of OA prevention and treatment a necessity to sustain the patients’ quality of life. In recent years, tissue engineering strategies for the regeneration of cartilage lesions have gained more and more momentum. In this study, we aimed to investigate the scaffold-free 3D cartilage tissue formation under simulated microgravity in the NASA-developed rotating wall vessel (RWV) bioreactor. For this purpose, we cultured both primary human chondrocytes as well as cells from the immortalized line C28/I2 for up to 14 days on the RWV and analyzed tissue morphology, development of apoptosis, and expression of cartilage-specific proteins and genes by histological staining, TUNEL-assays, immunohistochemical detection of collagen species, and quantitative real-time PCR, respectively. We observed spheroid formation in both cell types starting on day 3. After 14 days, constructs from C28/I2 cells had diameters of up to 5 mm, while primary chondrocyte spheroids were slightly smaller with 3 mm. Further inspection of the 14-day-old C28/I2 spheroids revealed a characteristic cartilage morphology with collagen-type 1, -type 2, and -type 10 positivity. Interestingly, these tissues were less susceptible to RWV-induced differential gene expression than those formed from primary chondrocytes, which showed significant changes in the regulation of *IL6*, *ACTB*, *TUBB*, *VIM*, *COL1A1*, *COL10A1*, *MMP1*, *MMP3*, *MMP13*, *ITGB1*, *LAMA1*, *RUNX3*, *SOX9,* and *CASP3* gene expression. These diverging findings might reflect the differences between primary and immortalized cells. Taken together, this study shows that simulated microgravity using the RWV bioreactor is suitable to engineer dense 3D cartilage-like tissue without addition of scaffolds or any other artificial materials. Both primary articular cells and the stable chondrocyte cell line C28/I2 formed 3D neocartilage when exposed for 14 days to an RWV.

## 1. Introduction

In 2019, about 528 million people worldwide were living with osteoarthritis (OA) [1]. This means an increase of 113.25% from 247.51 million patients in 1990 [1,2]. Long et al. reported an elevation in the prevalence of OA [2]. OA is a degenerative joint disease involving cartilage loss over time. It commonly occurs in the hands, knees, or hips but is also affecting other joints. Articular cartilage acts as a protective layer between the two bone ends of the joint; there it acts as a sort of shock absorber and ensures that the bones do not rub against each other. Cartilage is a smooth, avascular tissue and consists of the cartilage cells (chondrocytes) as well as the extracellular cartilage matrix. It is made up of proteins and carbohydrates. The tissue is bradytroph, because of the cartilage breakdown the patients suffer from pain, and their joints show an increase in stiffness and swelling. Therefore, prevention and an early start of therapy are important to mitigating the growing burden of this disease. The patients experience a progressive reduction of joint function. The current therapy strategies are as follows: Drug treatment with among others acetaminophen, or nonsteroidal anti-inflammatory drugs, cortisone injections, or surgery (joint replacements), physical therapy, lifestyle changes (exercise, weight reduction), and more [3,4].

Tissue engineering of cartilage and restoration of cartilage defects are currently hot topics in translational regenerative medicine. Known techniques are cell-based approaches, such as autologous chondrocyte implantation [5,6], matrix-induced autologous chondrocyte implantation [7], or tissue-engineered cartilage implantation [8,9]. Several bio-printing approaches and bioreactors for tissue engineering of cartilage are currently available and reviewed in [10]. Biofabrication or 3D bioprinting technology of functional cartilage can be applied in translational regenerative medicine or drug testing [11,12]. Importantly, 3D bioprinting is still in the early stages of development, it is costly, and comes with clinical and ethical challenges [13], but it provides the possibility to repair or replace damaged cartilage in OA patients. A big challenge in 3D bioprinting is the use of bioinks, because of the optimized properties necessary to engineer cartilage constructs that can be used in patients. Bioinks need to be insoluble in vivo and in cell cultures, have a stabile structure, promote cell growth, and they have to be non-toxic. Additionally, they need to interact with other cells in vivo. Bioinks can be damaged by the bio-printing process, which might result in reduced viability of the chondrocytes. Another drawback is that the 3D bioprinted cartilage tissue can induce a long-lasting immune response postimplantation due to the material of the bioink, infection, or toxicity [14].

An interesting and widely used bioreactor type is the National Aeronautics and Space Administration (NASA)-developed rotating wall vessel (RWV) bioreactor [15,16] which provides culture conditions mimicking aspects of microgravity. The RWV provides a reduction in shear stress and turbulences as compared to conventional stirred bioreactors like spinnerflask devices [17]. The device is effective for culturing primary cell lines in a more in vivo like environment [18]. The RWV bioreactor has to be equipped with a control system to vary the rotation speed of the vessel.

Microgravity presents a special environment for culturing normal or malignant cells and has been shown multiple times to induce cellular changes and processes, which could not be achieved on Earth and under static 1 *g* conditions. It has therefore gained increasing importance in the research field of tissue engineering, particularly since microgravity induces the scaffold-free formation of three-dimensional (3D) MCS or organoids [19].

Culturing cells under real and simulated microgravity resulted in changes in growth behavior and in the formation of 3D constructs or multicellular spheroids (MCS). Various cell types cultivated in space or on a microgravity simulating device revealed changes in adhesion, migration, differentiation, proliferation, growth, cytoskeleton, extracellular matrix, focal adhesion, secretion, among other biological processes [20]. Endothelial cells, tumor cells, and chondrocytes formed 3D aggregates in space [21,22,23] and on the RWV [24,25] or the random positioning machine (RPM) [20,26,27].

Chondrocytes cultured in simulated microgravity short-term showed various biological changes such as up-regulation of genes responsible for cell motility, structure, and integrity, as well as control of cell growth, proliferation, differentiation, programmed cell death, and cytoskeletal components [28]. Long-term exposure of human chondrocytes to the RPM resulted in the induction of 3D growth and the formation of cartilage [27] in the absence of scaffolds or micro-carriers. Then, 3D MCS were obtained after five days, and on day 28, the produced cartilage constructs reached a size up to 2 mm in diameter.

Earlier studies using bovine chondrocytes exposed to the RWV with Cytodex-3 microcarriers reported differentiated chondrocytes. That were observed in sections of RWV material after a time course of 36 days, while only a few were observed in the sections of control material [29]. Another publication demonstrated that primary chondrocytes grown on Cytopore-2 microcarriers maintained the phenotypical morphology and gene expression pattern observed in primary bovine articular chondrocytes, and retained these characteristics for up to 9 days [30]. The RWV as microgravity simulator has been proven to be suitable for cell culture without any scaffolds [31,32].

The principal aim of this study was to investigate the development of 3D cartilage-like tissue formation by both primary human chondrocytes and the immortalized cell line C28/I2 using the RWV in the absence of any scaffolds or artificial materials supporting tissue engineering cartilage. We intend to study whether microgravity-dependent cartilage formation is influenced by the features of a device which prevents cell sedimentation. After an exposure to the RWV up to 14 days, the C28/I2 cells were first macroscopically and microscopically investigated, and analyzed by histochemistry and immunohistochemistry to determine the amount and distribution of extracellular matrix components such as the collagen types I/II and X. Second, we turned our focus on changes in gene expression patterns of chondrocyte signaling factors and of key genes involved in cartilage production.

## 2. Materials and Methods

### 2.1. Cell Cultures

The commercially available immortalized human chondrocyte cell line C28/I2 (the cells were a kind gift from Mary Goldring) was cultured in complete medium (500 mL DMEM High Glucose (Sigma-Aldrich, St. Louis, MO, USA), FCS (10%), penicillin/streptomycin (1%), and sodium pyruvate (1%, 100 mM).

The cells from frozen stocks were grown in T175 cell culture flasks (175 cm^2^; Sarstedt, Nümbrecht, Germany) for 3 days. Afterwards, the cells were subcultured in three T175 cell culture flasks for RWV experiments and further subculturing. The cells were cultured in a commercial incubator at 37 °C and 5% CO_2_ until confluency. Then, the cells were trypsinized, centrifuged and the pellet was then resuspended in 10 mL of fresh medium. 10^7^ cells from the 3rd subculture were transferred in suspension to each RWV culture vessel. The other part of the cells was cultured as static control group in T175 flasks (10^7^ cells per flask).

A similar cell culture protocol was applied for primary human chondrocytes (HCHON, Provitro, Berlin, Germany). We used Chondrocyte Growth Medium (CGM; Provitro^®^, Berlin, Germany), supplemented with 10% fetal calf serum (Biochrom^®^, Berlin, Germany) and antibiotics (100 IU penicillin/mL and 100 µg streptomycin/mL, Biochrom^®^, Berlin, Germany) [27]. After reaching passages 3–4, 5 × 10^6^ cells per vessel were transferred into two RWV vessels and 0.15 × 10^6^ cells per flask into five T75 flasks for control. For the second run, we used 12 × 10^6^ cells (passage 3–4) per vessel in two RWV vessels and again 0.15 × 10^6^ cells per flask in five T75 flasks for control.

### 2.2. Rotating Wall Vessel Device

We used an RCCS-2D system with two 50 mL disposable vessels (Synthecon Incorporated, Houston, TX, USA) (Figure 1). Suspensions of cultured cells were inoculated into each 50 mL high-aspect rotating culture vessel (HARV) with permeable membrane that were subsequently completely filled with medium, carefully avoiding air bubbles. The RWV was placed into an incubator at 37 °C and 5% CO_2_. The rotary culture with the C28/I2 cell line was performed for 3, 7, and 14 days. The rotation speed was first adjusted to 13 rounds per minute (rpm), increasing the speed to 20 rpm after the first 48 h. The chondrocyte medium was not changed for experiment durations of 7 days or less.

The primary human articular chondrocyte (HCHON) cells were exposed to the RWV for 14 days. Then, 5 × 10^6^ cells were seeded in one vessel which was filled air-bubble-free with complete culture medium. The device started at 13 rpm and increased its speed to 14.8 rpm during a week. After a 7-day RWV-exposure, the culture medium was exchanged.

For the normal gravity control groups for the C28/I2 experiments, chondrocytes of the same passage were seeded into T175 cell culture flasks (5 × 10^6^ cells per flask) with 50 mL culture medium. For the HCHON 1 *g* control group, 0.15 × 10^6^ cells of the same passage were seeded into 5 T175 flasks.

### 2.3. Histological Analysis

Half of the human chondrocyte spheroids obtained after the 3-, 7-, and 14-day RWV exposition were fixed in 4% paraformaldehyde, embedded in paraffin, and cut into 4 µm sections. The sections were mounted onto slides and dried. The other spheroids were used for RNA extraction and qPCR.

#### 2.3.1. Histological Staining

Histological sections were deparaffinized and stained according to protocols using hematoxylin-eosin, Sirius red, and Safranin-O/FastGreen staining [33,34].

#### 2.3.2. Immunohistochemistry

Histological sections were deparaffinized and subsequently demasked using trypsin/EDTA (Biochrom, Berlin Germany) for 10 min at 37 °C. Blocking was then performed in 4% bovine serum albumin fraction V (BSA; Sigma-Aldrich, St. Louis, MO, USA). All sections were prepared with primary antibodies against collagen 1 (Anti-COL1A1, Novus Biologicals, Centennial, CO, USA; 1:500), collagen 2 (Anti-COL2A1 H-300, Santa Cruz Biotechnology, Dallas, TX, USA; 1:100), and collagen 10 (Anti-COL10A1, Abcam, Cambridge, UK; 1:150) and incubated overnight at 4 °C. The sections were then incubated with the fluorescence-labeled secondary antibody anti-rabbit Alexa 555 (1:200, Invitrogen, Carlsbad, CA, USA) for 60 min at room temperature. Sections were covered with ROTI-Mount FluorCare DAPI (Carl Roth GmbH u. Co. KG, Karlsruhe, Germany) and microscopically investigated (Axio Observer, Axiocam 702 mono, HXP 120 V, Carl Zeiss, Jena, Germany). All sections were controlled for unspecific binding by incubation with an iso-rabbit antibody (Invitrogen, Waltham, MA, USA) [35].

### 2.4. Cell Death Detection

For apoptosis detection the histological sections were deparaffinized and washed in PBS (Biochrom, Berlin Germany). The chondrocytes were stained using the In Situ Cell Death TMR kit (Merck, Darmstadt, Germany) following the manufacturer’s instructions. The sections were covered with ROTI-Mount FlourCare DAPI (Carl Roth GmbH u. Co. KG, Karlsruhe, Germany). Afterwards, the fluorescence images were analyzed by cell counting using ImageJ for Windows. TUNEL- and DAPI-positive cells were put into relation ((TUNEL positive cells/total number of cells) *100) [36].

### 2.5. RNA Isolation

C28/I2 spheroids were transferred in RNA stabilizing reagent (RNA*later*, Ambion), snap frozen, pulverized in liquid nitrogen and lysed with Trizol (Invitrogen, Carlsbad, CA, USA) following the manufacturer’s instruction for total RNA isolation. RNA was loaded on Nanoquant plate (TECAN Deutschland, Crailsheim, Germany) to determine RNA concentration using Infinite F200 Pro (TECAN Deutschland, Crailsheim, Germany). A total of 1 µg RNA was reverse transcribed using high capacity cDNA reverse transcription Kit (Thermo Scientific, Wilmington, DE, USA) and T100 Thermal cycler (BioRad, Shinagawa, Tokyo) following manufacturer´s instruction.

For RNA extraction of the primary chondrocytes, half of the obtained human spheroids were collected in 50 mL tubes and fixed in RNA*later*TM (Invitrogen, Waltham, MA, USA). Static control cells were scraped off using cell scrapers (Sarstedt, Nümbrecht, Germany), and were also transferred to 50 mL tubes, fixed and pelleted by centrifugation (2500× *g*, 10 min, 4 °C). We used a RNeasy Mini Kit (Qiagen, Hilden, Germany). According to the manufacturer’s instructions, we isolated the total RNA. Concentrations and quality of the RNA were evaluated spectrophotometrically at 260 nm using a NanoDrop instrument (Thermo Scientific, Wilmington, DE, USA). The isolated RNA had an A260/280 ratio of 1.5 or higher. cDNA to perform quantitative real-time PCR was then procreated utilizing the First-Strand cDNA Synthesis Kit (Fermentas, St. Leon-Rot, Germany) using 1 µg of total RNA in a 20 µL reaction mixture at room temperature.

### 2.6. Quantitative Real Time PCR (qPCR)

qPCR was used to examine the mRNA expression of selected genes after a 3-, 7-, and 14-day RWV exposure of human C28/I2 chondrocytes, as well as after a 14-day RWV exposure of HCHON cells and compared to the static 1 *g* cell-line specific control groups. The method was previously described in detail [37].

All samples were run on an Applied Biosystems QuantStudio 3 PCR system using the Fast SYBR^®^ Green PCR Master Mix (both Applied Biosystems, Darmstadt, Germany). The reaction volume was 15 μL including 1 μL of template cDNA and a final random hexamer primer concentration of 300 nM. The qPCR conditions were as follows: 20 s at 95 °C, 40 cycles of 1 s at 95 °C, and 20 s at 60 °C, followed by a melting curve analysis step (temperature gradient from 60 to 95 °C with +0.3 °C/cycle). Every sample was measured three times. The comparative C_T_ (ΔΔC_T_) method was used to relatively quantify the transcription levels. 18S rRNA was used as a housekeeping transcript to normalize the expression data. Before performing qPCR, all primers were designed using NCBI Primer Blast, and they were selective for cDNA by spanning exon–exon junctions and had a melting temperature of around 60 °C. The primers were synthesized by TIB Molbiol (Berlin, Germany) and are listed in Table 1.

### 2.7. STRING Analysis (Search Tool for the Retrieval of Interacting Genes/Proteins, V11.5)

The interactions between genes/proteins were analysed using the STRING 12-0 platform [38] (available at https://string-db.org/, accessed on 28 November 2023). For each protein, the UniProtKB entry number was inserted in the input form “multiple proteins,” and “Homo sapiens” was selected as the organism.

### 2.8. Statistical Analysis

The statistical analyses were performed using IBM SPSS Statistics 23 (IBM Deutschland GmbH, Ehningen, Germany). The Mann–Whitney U-Test was utilized to evaluate the statistical significance in the changes in expression levels following RVW exposure, thus comparing the static 1 *g* control to microgravity samples (RWV samples, cartilage constructs, neocartilage). A significance level of 0.05 was used. The standard deviation was calculated and presented together with the mean values as percentages in bar plots. Where necessary, separate runs were combined with the Cochrane method and subsequently tested for statistical differences with the *t*-test.

## 3. Results

### 3.1. Formation of 3D Cartilage Constructs in the Rotating Wall Vessel

The RWV was used to simulate microgravity in order to engineer 3D aggregates without the addition of scaffolds or other materials (Figure 1). Primary human chondrocytes (passage 3) purchased from Provitro ^®^, Berlin, Germany, were randomized to controls for 14-day experiments as static 1 *g* cultures (Figure 1C) and for investigation on the RWV (Figure 1B,D). Figure 1C represents normal human articular chondrocytes undifferentiated at 100% confluency after 14 days (1 *g* cultures), whereas RWV-cultures grew as 3D aggregates. After 14 days, the vessels revealed 3–5 3D tissues of a maximal size of 4 mm (white mark, Figure 1D).

Furthermore, we studied the C28/I2 human chondrocyte cell line, a widely used model cell line to focus on cartilage repair mechanisms. The C28/I2 chondrocytes exposed to the RWV started to form visible small 3D aggregates already after 3 days of continuous cultivation (Figure 1E). Various round tissues were visible as indicated by white arrows. On the 7th day of RWV exposure, the visible aggregates were about 2 mm in size (Figure 1F). The maximal size of 5 mm in diameter was achieved after 14 days of cultivation (Figure 1H). The macroscopic examination of the 3D aggregates revealed a round morphology after 7d of continuous RWV-exposure, characterized by the compact tissue mass and dense matrix production.

### 3.2. Histochemical and Immunohistochemical Investigation of 3D C28/I2 Tissues

The HE-stained paraffin sections of the 3D tissues exhibited a cell-rich organization of cartilaginous tissue. This was observed in 7-and 14-day-old aggregates (Figure 2A,B). Moreover, individual cells forming cartilaginous matrix (reddish orange—purple color) were clearly visible in safranin-O/fast green stained spheroids (Figure 2C,D). In addition, during the incubation of chondrocytes on the RWV, we observed an enhanced collagen production which was shown by Sirius red staining (Figure 2E,F).

Collagen type I was detectable in the interstitial space of 3-day-old spheroids, but its amount was elevated by the time of 14 days of RWV-exposure (Figure 3A–C). Vice versa, little collagen type II was detectable at the 3rd and 7th day of RWV-exposure, but it was increased in 14-day-old samples (Figure 3D–F).

Positive staining for human collagen type 10 was observed in the same areas as that for collagen type I (Figure 3G,H), and in addition, a clearly positive collagen type 10 protein content was detected after 7 and 14 days (yellow arrows; Figure 3H,I). TUNEL stains of 3-, 7-, and 14-day-old spheroids exhibited a moderate time-dependent increase in apoptosis (Figure 4A–C).

### 3.3. Gene Expression of Chondrocyte Signaling Factors in 3D C28/I2 Spheroids

Quantitative real-time PCR (qPCR) did not reveal any changes in the gene expression of *COL1A1*, *COL2A1*, *COL10A1*, *SPP1*, *PRG4*, *SOX9*, *IL6*, *CXCL8*, *TGFB1*, *RUNX2*, *CASP3*, *CASP8,* and *MMP13* in 3-day samples, irrespective of the culture condition (Figure 5A–M). Significant downregulations of genes after a 7-day RWV exposure were measured for the following mRNAs in RWV samples compared to static 1 *g*: *COL1A1*, *COL1A2*, *SOX9*, *IL6*, *TGFB1*, *RUNX2,* and *CASP3* (Figure 5A,B,F,G,I–K). Significant changes in the gene expression pattern after a 14-day RWV-exposure were measured in 3D aggregates as follows: A significant upregulation of *SOX9*, *CXCL8*, and *TGFB1* mRNAs was measured in 14-day RWV aggregates compared to 7-day RWV aggregates (Figure 5F,H,I) and the *CASP3* mRNA was downregulated in 7-day RWV tissues compared to 3-day RWV tissues (Figure 5K).

### 3.4. Gene Expression Pattern in Primary Human Chondrocyte Spheroids

In a second step, we investigated tissue engineered 14-day-old RWV cartilage pieces (neocartilage, spheroids) obtained from primary chondrocytes. The qPCR revealed significant downregulations in the following genes: *ACTB*, *TUBB*, *VIM*, *LAMA1*, *ITGB1*, and *COL1A1* in 14-day cartilage tissues compared to static 1 *g* samples (Figure 6A–E,H).

In addition, a significant upregulation of *COL2A1*, *COL10A1*, *MMP1*, *MMP3*, *RUNX3*, *IL6*, *SOX9,* and *CASP3* was measured in 14-day spheroids compared to static 1 *g* samples (Figure 6F,G,K,L,N,R,T,U).

The qPCR revealed no changes in gene expression for *ACAN*, *SPP1*, *RUNX2*, *TGFB1*, *SOX6*, *SOD3*, *CXCL8,* and *CASP8* (Figure 6I,J,M,O–Q,S,V).

### 3.5. Search Tool for the Retrieval of Interacting Genes/Proteins (STRING) Analysis

Figure 7A shows a summary of the qPCR data given in Figure 5 and presents an interpretation of the results. *IL6, COL1A1, COL2A1, TGFB1, SOX9, RUNX2, CASP3,* and *CASP8* were downregulated in 7-day neocartilage engineered on the RWV. 3-day and 14-day RWV cultures showed no differences in the gene expression pattern *(COL1A1, COL2A1, COL10A1, SPP1, PRG4, MMP3, SOX9, IL6, CXCL8, TGFB1, RUNX2, CASP3*, and *CASP8*) compared to corresponding static 1 *g* cultures.

Figure 7B visualized the network of the functional interaction of genes and proteins. It demonstrates the interactions of 13 genes of interest investigated in this study. Eight genes were downregulated in neocartilage vs. corresponding 1 *g* samples at day 7.

Figure 7C gives the qPCR determined fold changes of several genes (*IL6, CXCL8, ACTB, TUBB, VIM, ACAN, COL1A1, COL2A1, COL10A1, ITGB1, LAMA1, MMP1, MMP13, SPP1, PRG4, RUNX2, RUNX3, TGFB1, SOX6, SOX9, SOD3, CASP3, CASP8*) involved in cartilage tissue engineering. The cytoskeletal and ECM genes as well as SOX transcription factors had already proved their gravisensitivity in earlier studies in the field of space research [39,40].

Figure 7D documents the interactions of all genes given in Figure 7C. The results of the analysis indicate several interactions for the following interesting target genes: *MMP1, MMP13, SPP1, ITGB1, ACAN, IL6, RUNX3*, and *SOX9*.

## 4. Discussion

Gravitational biology, space medicine, cancer research in space, and tissue engineering, as well as the biofabrication of organoids, are currently hot topics in the field of space research [19,20,41].

In the context of space exploration, it is necessary to expand our knowledge about the effects induced in human cells by stressors like microgravity. The application of advanced technologies such as 3D printing, tissue engineering, and regeneration among others plays a key role in the implementation of future research programs recommended by NASA, ESA (European Space Agency), and other space agencies [42].

The unique feature of our study is the s-µ*g*-based analysis of biomarkers of cartilage biosynthesis. Markers of cartilage biosynthesis have been sufficiently described in previous studies, but the knowledge of the µ*g* influence on this biosynthesis is crucial for the development of complex in vitro 3D cartilage tissues, which, in future, will play a decisive role in the treatment of cartilage damage and osteoarthritis on extended space missions and in personalized medicine on Earth. Through the final in silico linkage of differential key genes (Figure 7), we were able to identify and map key gene interactions in 3D C28/I2 spheroids as well as in primary human chondrocyte spheroids.

For a long time, it has been known that cells cultured under conditions of real and simulated microgravity exhibit various structural changes. Inter alia, benign, and malignant cells exposed to microgravity show changes in cell adhesion, proliferation, apoptosis, migration, and growth behavior [43]. Three-dimensional growth of different kinds of tumor cells and healthy cells cultured under microgravity was already described in the early 1990s and 2001 [44,45,46,47,48,49].

### 4.1. Tissue Engineering under Microgravity Conditions

The research status of growing tissues in microgravity was reviewed by Unsworth and Lelkes in 1998 [44]. Engineering of 3D aggregates under real and simulated microgravity in orbit or on Earth in our laboratories provides important insights for future space exploration with populations living in Mars and Moon villages. It provides humans the possibility to create new organoids, or organs for transplantation in space or researchers to perform pharmacological, toxicological, radiation tests, or experiments in the field of cancer research [19,20].

The NASA-developed RWV bioreactor was, and still is, widely used to study cancer cells under microgravity [50,51]. Other examples for usages were published by Freed et al. in 1993, and by Marlovits et al. Ten years later in 2003, both focusing on chondrocytes with and without scaffolds exposed to the RWV [31,32,48].

In our study, primary human chondrocytes and chondrocytes from the stable C28/I2 cell line were exposed to an RWV, which generated dense 3D cartilage-like tissue in the absence of scaffolds or any other artificial materials. The RWV bioreactor prevents cell sedimentation at low shear force, is widely used for tissue engineering purposes and has been proven to be suitable to trigger chondrocytes to grow three-dimensionally and to form cartilage-like tissues [31,32]. Using the RWV, the process of aggregation already started after 3 days of continuous cultivation of C28/I2 chondrocytes, and these tissues reached a size of about 5 mm after 14 days of cultivation (Figure 1). In contrast, cartilage tissue obtained from primary chondrocytes after a 14-day RWV-exposure exhibited a size of 3 mm in diameter.

Another device simulating microgravity is the RPM, which also has demonstrated to be suitable to engineer cartilage tissue [27]. Chondrocytes (HCHON, Provitro^®^) exposed to the RPM grew in form of 3D spheroids after five days without addition of any scaffolds. On day 14, the produced tissue constructs cultured in specific chondrocyte medium reached up to 2 mm in diameter and exhibited a typical cartilage morphology [27].

Moreover, the RPM was used to investigate early effects of microgravity on human chondrocytes [28]. Short-term microgravity periods provided by parabolic flight maneuvers influenced human chondrocytes to alter their cytoskeleton moderately. This change was accompanied by an up-regulation of BMP-2, TGF-β1, and SOX9 [39]. Interestingly, in our long-term experiments we only observed an upregulation of *SOX9* gene expression, while *TGFB1* remained essentially unchanged. Moreover, the cellular viability was not affected by parabolic flight maneuvers [40]. This result was supported by a significant upregulation of anti-apoptotic genes measured after 31 parabolas [40]. The microgravity-exposed chondrocytes altered the synthesis of extracellular matrix proteins, and rearranged their cytoskeleton prior formation of 3D aggregates [28].

Both microgravity simulation devices have been proven to be suitable for cartilage tissue engineering. Earlier studies had reported the effects of real microgravity in space. Freed et al. were the first who investigated cell-polymer constructs consisting of bovine articular chondrocytes on polyglycolic acid scaffolds on the MIR space station [48]. The authors first cultured chondrocytes for 3 months on Earth and subsequently for 4 months on the MIR, and in parallel on Earth in a bioreactor, producing 3D cartilaginous aggregates. The tissues exhibited viable, differentiated cells synthesizing proteoglycan, and collagen type 2 [48]. Another study was performed using the RPM and porcine chondrocytes by Cogoli et al. in the 2000s [23,49]. Porcine cartilage cells were cultured during the Flight 7S (Cervantes mission) on the International Space Station (ISS), in simulated microgravity using an RPM, and under normal 1 *g* control conditions. After 16 days, the neocartilage tissue exhibited a weaker extracellular matrix in ISS samples compared to both Earth tissues. Higher collagen II/I expression ratios were measured in ISS cartilage tissue compared to 1 *g* controls. Furthermore, the authors reported higher aggrecan/versican gene expression profiles in 1 *g* control samples compared to microgravity samples. The authors concluded that application of the RPM may be a useful tool to produce cartilaginous tissue grafts [23].

### 4.2. Extracellular Matrix Proteins, Cytokines and Chondrocyte Signaling Factors in Neocartilage Tissue-Engineered with the Stable C28/12 Chondrocyte Cell Line

RWV engineered 3D tissues show a characteristic cartilage morphology with collagen-type 1, -type 2, and type 10 positivity. This was supported by histochemical staining with safranin-O, sirius red and hematoxylin-eosin (Figure 2). These spheroids show properties making them interesting tissues for cellular cartilage regeneration approaches in trauma and OA patients.

The gene expression of *COL1A1* was downregulated in RWV tissues compared to static controls but remained unchanged at a low level after 14 days. In parallel, the *COL2A1* mRNA was decreased in 7-day samples compared to static 1 *g* controls but was clearly measurable at all time points. The amount of *COL2A1* mRNA was similar in 3-, 7-, and 14-day RWV tissues. Immunohistochemical investigation of the collagens type 1, and type 2 documented the presence of both proteins especially in 14-day tissues (Figure 3). Furthermore, the *COL10A1* mRNA was expressed in all samples but was not significantly altered at all time points. A similar result was found in chondrocytes exposed for 24 h on the RPM [27]. COL10A1 is a marker for hypertrophic chondrocyte differentiation and is regulated by RUNX2 [50,51]. The *RUNX2* gene expression was not significantly changed after 3 days and 14 days but was downregulated after a 7-day-exposure in RWV tissues compared to static 1 *g* controls (Figure 5J). This finding might explain why *COL10A1* remained unchanged. A similar result was found for *SPP1* which is also regulated by RUNX2 [51]. The *SPP1* gene was not differentially regulated in RWV tissues, compared to corresponding static 1 *g* controls. In addition, the *PRG4* (proteoglycan 4) gene expression was not changed at any time points in the RWV tissue vs. the static 1 *g* control group (Figure 5E). PRG4 is beneficial for cartilage due to its osmotic properties, supporting articular cartilage to resist compressive loads. The secreted proteoglycan lubricin is expressed by superficial zone articular chondrocytes and shown to be sensitive to mechanical loading and is cell-protective against the development of OA [52]. Moreover, the matrix metalloproteinase-13 (*MMP13*) mRNA was not altered at any time points in our experimental setting [Figure 5M]. MMP-13 is the primary MMP involved in cartilage degradation which can be explained by its role in cleaving type II collagen [53]. Because of its role in degradation, MMP13 might be an interesting target for OA treatment [53].

The *IL6* gene expression was low in RWV tissues at all time points. A significant downregulation of the *IL6* mRNA was detected after 7 days in RWV samples vs. static 1 *g* controls. Il-6 plays a key role in OA development and increased serum levels of this proinflammatory cytokine correlate with disease severity and are responsible for the induction of matrix-degrading enzymes [54]. Of interest is the dual role of IL-6, because it also acts protective through elevating anti-catabolic factors [54]. We also focused on interleukin-8 (*CXCL8*). The *CXCL8* gene expression was not differentially displayed RVW samples compared to the corresponding 1 *g* controls. Interestingly, in 14-day samples the amount of mRNA was higher than after 7 and 3 days proposing a possible role of CXCL8 in differentiation and maturation of a cartilage-like structure in vitro and consequently the regulation of cartilage homeostasis under microgravity conditions.

In a next step, we focused on the SRY-box transcription factor 9 (SOX9) which is involved in chondrogenesis and chondrocyte differentiation by promoting the transcription of its target genes. *SOX9* was low after 3 days in 1 *g* and RWV samples, and significantly elevated in 7-day 1 *g* cultures. The *SOX9* gene was downregulated in 7-day RWV samples compared to corresponding 1 *g* samples and elevated in 14-day RWV samples vs. 7-day RWV samples (Figure 5F), indicating its involvement in cartilage tissue engineering. The *SOX9* expression is downregulated in cartilage of OA patients but elevated during tumorigenesis of cartilage and bone. SOX9, SOX6, and SOX5 application into cells supports chondrogenesis [55]. Various signaling pathways influence the SOX9 expression during chondrogenesis [55]. One of these signaling factors is *TGFB1*, which was also significantly downregulated in 7-day RWV tissues compared to corresponding 1 *g* controls, and significantly enhanced in 14-day RWV samples compared to 7-day RWV tissues (Figure 5I). This finding fits well to previous reports found in the literature and shows that TGFB1 and SOX9 have a similar chondroprotective function for 3D growth of chondrocytes exposed to an RWV.

### 4.3. Signaling Factors of Different Biological Processes in Neocartilage Tissue-Engineered with Primary Human Articular Chondrocytes

We studied the effects of simulated microgravity on commercially available primary chondrocytes of low passages and focused on 3D growth and the engineering of neocartilage. It is of interest to evaluate differences in growth behavior of primary cells and the stable chondrocyte cell line C28/I2.

It has been shown that after the 30-day BION-M1 space mission articular cartilage of mice revealed signs of degradation at the tissue level, with loss of proteoglycan as well as a downregulated mRNA expression of mechano-responsive and structural cartilage matrix proteins compared to ground controls [56]. The authors demonstrated that sternal cartilage responded differently to the spaceflight environment [56].

Significant downregulations in cytoskeletal components were measured by qPCR in chondrocytes exposed to the RVW. The gene expression of *ACTB*, *TUBB,* and *VIM* was clearly reduced in 14-day RWV tissues compared to static controls (Figure 6A–C). This is in contrast to short-term data, obtained from parabolic flight missions. The mRNA expression and protein content of primary articular chondrocytes exposed to the 31st parabola of a parabolic flight mission revealed upregulated cytoskeletal genes and proteins (*TUBB*, 2-fold; *VIM*, 1.3-fold; *ACTB*, 1.9-fold) [39]. These results document the early role of the cytoskeleton as gravi-sensor. Chondrocytes exposed to the RPM for up to 24 h changed the ECM production while they rearranged the cytoskeleton prior to forming 3D tissues [28]. A decrease in the amount of *LAMA1*, *COL1A1,* and *ITGB1* mRNAs was accompanied by an increase in the COL2A1 and COL10A1 mRNA expression in RVW cartilage fragments compared to 1 *g* samples. Moreover, *ACAN*, *SPP1,* and *PRG4* genes were not differentially expressed in 14-day tissues. These results show that primary chondrocytes exposed to the RWV remain well-differentiated. The low *COL1A1* and high *COL2A1* gene expression supports this thesis. It is known, that a normal articular cartilage function needs a high amount of aggrecan, a high degree of aggrecan sulfation, and the ability to form large aggregates [57]. Therefore, a normal *ACAN* expression in the RWV tissues is of advantage for a normal function of the neocartilage tissues. In addition, the normal amount of *PRG4* mRNA in the neocartilage indicates that the tissues are well protected against external stressors and diseases [58]. The expression of osteopontin is involved in chondrocyte hypertrophic differentiation in OA disease and progression [59]. *SPP1* (secreted phosphoprotein 1) was upregulated in chondrocytes after 24 h RPM exposure, which indicates its role in cell detachment and following aggregation [27]. SPP1 is a secreted ECM protein, that binds to cell surface integrins inducing cell–cell and cell–ECM adhesion and communication, which are early biological processes.

Furthermore, the gene expression of *MMP1*, *MMP3,* and *MMP13* was significantly elevated in RWV tissues compared to 1 *g* samples. Matrix metalloproteinases are contributing to the degenerative process occurring during OA pathogenesis. MMPs are involved in ECM remodeling. These endopeptidases are able to degrade the ECM components [60]. MMP1 and MMP13 belong to the collagenases and MMP3 to the stromelysins [60]. The collagenases degrade collagens Type II, II, III, VIII, X, gelatins, aggrecan, and entactin, whereas stromelysins among others degrade proteoglycans, laminin, fibronectin, entactin, collagens type III, IV, V, IX, X, and XI [60]. Their upregulation might explain the decrease in *COL1A1*.

Runt-related transcription factors (Runx) are involved in the homeostasis of cartilage. Runx3 acts as cartilage-protective through ECM protein production under normal conditions, while Runx2 exhibits both catabolic and anabolic effects in inflammation [61]. Interestingly, *RUNX3* was significantly elevated in 14-day neocartilage samples, whereas *RUNX2* remained unaltered compared to the 1 *g* condition.

*SOX6* remained unaltered, but *SOX9* was significantly elevated in RVW tissues compared to 1 *g* controls. This finding supports the important role of SOX9 in chondrogenesis and cell-protective role in microgravity [39,55].

The increase in IL6 is paralleled by an elevation of the *CASP3* gene expression in RVW samples. Healthy cartilage tissue may produce small quantities of IL-6 [62]. Highly upregulated IL-6 can induce MMP-3 and MMP-13 production and promote cartilage ECM degradation, which may be paralleled by apoptosis.

### 4.4. Interaction Network of Selected Genes Evaluated by STRING Analysis

#### 4.4.1. C28/I2 Chondrocyte Cells Exposed to Microgravity Conditions

The STRING analysis showed various interactions between the studied genes of interest, with the majority of them demonstrating many interactions. Among others, *SOX9*, *COL1A1*, *IL6,* and *TGFB1* are indicated as being dominant target genes, as many arrows point to their icons (Figure 7B). SOX9 is an SRY-related transcription factor involved in the expression of cartilaginous genes during development. SOX9 interacts with *COL1A1, COL2A1, COL10A1, SPP1, PRG4, MMP13,* and *RUNX2*. SOX9 induces the expression of PG4 and COL2A1, which was demonstrated by *SOX9* gene transfer. The proteoglycans and type II collagen expressions were upregulated over time in normal and OA articular chondrocytes in vitro [63]. The transcriptional regulation of *COL2A1* and *MMP13* by RUNX2 and SOX9 was investigated by Nagata and team [61]. The authors showed that RUNX2 enhanced *MMP13*, whereas it compensates for reduced *COL2A1* expression in articular chondrocytes under inflammatory conditions in which *SOX9* expression is decreased. SOX9 is interacting with *SPP1* transcriptionally through downregulation of this gene in order to prevent matrix mineralization in chondrocytes and heart valves [64]. The authors presented that SPP1 is important for matrix mineralization induced by SOX9 knockdown [64].

Another target gene in chondrocyte signaling is *COL1A1,* which revealed interactions with *RUNX2*, *SOX9*, *MMP13*, *CASP8*, *COL2A1,* and *SPP1*. RUNX2 is known to regulate the differentiation of chondrocytes and the expression of various ECM genes. For example, it regulates the *COL10A1* expression in hypertrophic chondrocytes and the expression of *SPP1*, *IBSP*, and *MMP13* in terminal hypertrophic chondrocytes. During osteoblast differentiation, RUNX2 upregulates the expression of bone matrix protein genes, including *COL1A1*, *SPP1*, or *FN1* in vitro [51].

*IL6* interacts with *MMP13, CASP8, CASP3,* and *CXCL8*. In addition, *TGFB1* exhibits interactions with *SPP1, COL2A1, CASP8,* and *CXCL8*. Pro-inflammatory cytokines like IL-6 are able to induce apoptosis in chondrocytes [65]. Apoptosis occurs in OA cartilage, but it is still not clarified whether chondrocyte apoptosis is the inducer of cartilage degeneration or a byproduct of cartilage destruction. The increase of caspase-8 and caspase-3 induced by inflammatory cytokines and mechanical stress, among other factors, results in DNA fragmentation and induces MMPs, but reduces collagen type 2 and aggrecan in chondrocytes [66]. Moreover, SOX9 regulates cell survival and suppresses apoptosis in limb bud mesenchyme [67].

#### 4.4.2. Primary Articular Chondrocytes Exposed to the Rotating Wall Vessel

The interaction analysis demonstrated a large number of interactions between the studied chondrocyte signaling genes. Inter alia, *IL6*, *COL2A1*, *COL10A1*, *MMP3*, *MMP13*, *RUNX3*, *SOX9*, and *CASP3* are indicated as being dominant target genes because various arrows point to their icons (Figure 7D). The studied genes of interests belong to the following biological processes and factors playing a key role in skeletal signaling and development: Cytokines, cytoskeleton, extracellular matrix, cartilage turnover, cartilage surface, osteoblast differentiation, Wnt signaling, growth factors, chondrogenesis, antioxidants, and apoptosis. The various, for cartilage tissue engineering critical, interactions have been discussed in Section 4.4.1.

Interestingly, the gene expression pattern of 14-day neocartilage fragments from C28/I2 cells was not differentially regulated compared to corresponding controls. This is in contrast to the results obtained with primary articular chondrocytes. As summarized in Figure 7C, significant fold changes were determined for the following genes: *IL6*, *ACTB*, *TUBB*, *VIM*, *COL1A1*, *COL10A1*, *MMP1*, *MMP3*, *MMP13*, *ITGB1*, *LAMA1*, *RUNX3*, *SOX9,* and *CASP3*. This difference might be explained by the fact that the primary articular chondrocytes were used in a low passage, whereas the C28/12 cell line is a stable cell line in high passages. Moreover, the C28/I2 cells were established from costal cartilage and the primary chondrocytes were isolated from hip joint cartilage. Furthermore, the C28/I2 cells were immortalized with the SV40 large T antigen inserted in the retroviral neomycin-resistant pZipNeoSV(X) vector and transfected using polybrene [68].

In male Sprague Dawley rats with immobilized right limbs, an increase in MMP-3 levels was observed after only 6 h. This effect was reversed after only one hour of joint mobilization [69]. Additionally, after 30 days of Hindlimb unloading (HLU) in C57BL/6 mice a significant increase of MMP-13 concentration could be noted [70]. In contrast the Mmp13*,* Mmp9 and Col1a1 gene expression is markedly decreased in sternal cartilage of C57BL/6 mice after 30 days of r-µg exposure in the Bion-M1 space mission [56]. In osteoarthritic rats ITGB1 overexpression activates the cAMP pathway with consequences in the regulation of inflammation [71]. ELSIA measurements of TNFa, IL-1a/β, IL-6, and IL-10 in male Sprague Dawley rats with immobilized glenohumeral joints of the shoulder showed a significant inflammation response and an increased level of the five cytokines [72]. In summary, compared to our study, we find an overlap of gene and protein regulation in the animal model under the influence of demobilization and microgravity.

Taken together, this study showed that simulated microgravity using the RWV is suitable to engineer dense 3D cartilage-like tissues without addition of scaffolds or any other artificial materials. Both primary articular cells and a stable chondrocyte cell line formed 3D neocartilage when exposed to an RWV for 14 days.

## 5. Conclusions

Tissue engineering of cartilage under simulated microgravity is an interesting method for translational regenerative medicine [73,74]. In the view of future planned extensive space exploration to the Moon and Mars and the aging population on Earth, the development of new promising techniques and innovations to repair cartilage defects with small neocartilage after injuries or to substitute cartilage in the joints of OA patients is absolute necessary. The RWV technology provided growing viable tissue during the 14-day culture with both cell types and therefore might be a useful technique for chondrocyte amplification in vitro.

## Figures and Tables

**Figure 1 biomolecules-14-00025-f001:**
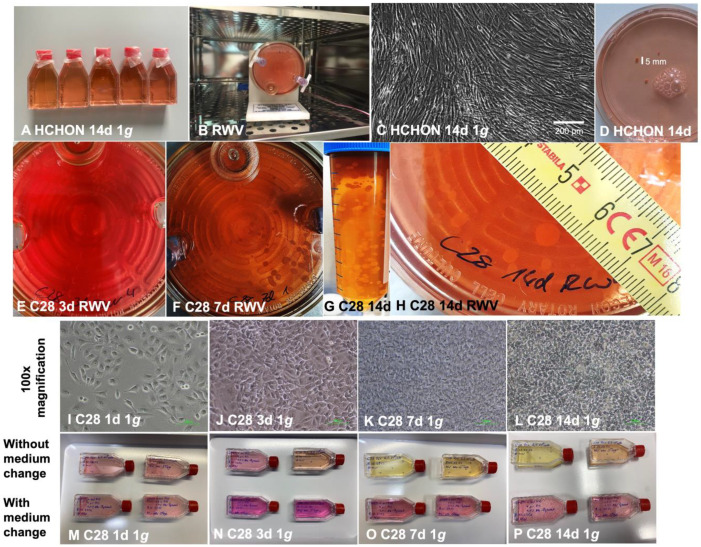
Experimental setup and resulting spheroids from chondrocyte cultivation under simulated microgravity conditions in the Rotating Wall Vessel bioreactor: (**A**): static 1 *g* HCHON control culture flasks. (**B**): RWV positioned in an incubator at 37 °C and 5% CO_2_. (**C**): Primary human chondrocytes (HCHON) cultured for 14 days under static 1 *g* conditions. (**D**): After 14 days on the RWV, HCHON had formed 3D spheroids of macroscopic scale. (**E**–**H**): Time course of spheroid development formed from C28/I2 cells grown on the RWV for 3, 7, and 14 days, respectively. At the end of the experiment spheroids with diameters of up to 5 mm could be observed. (**I**–**L**): Phase contrast microscopy of 1d, 3d, 7d and 14d control C28/I2 chondrocytes exposed to 1 *g* (normal gravity) conditions. (**M**–**P**): static 1 *g* C28/I2 control culture flasks (top: without medium change and bottom: with medium change).

**Figure 2 biomolecules-14-00025-f002:**
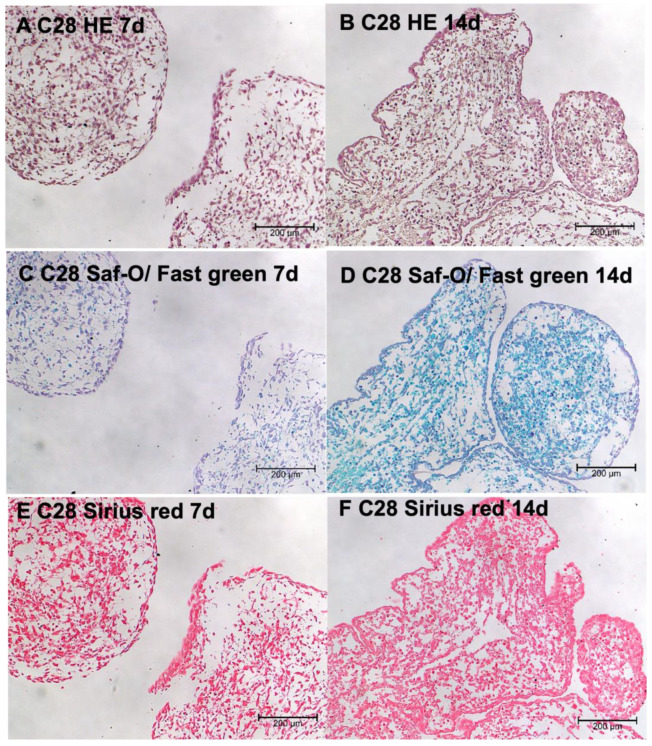
Representative images of spheroid sections stained with HE (**A**,**B**), safranin O/fast green (**C**,**D**), and sirius red (**E**,**F**) staining of spheroids from C28/I2 cells grown on the RWV after 7 and 14 days.

**Figure 3 biomolecules-14-00025-f003:**
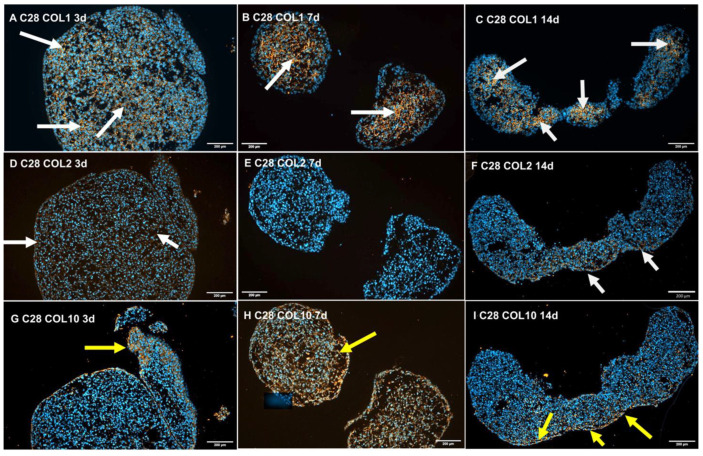
Immunofluorescence staining for collagen type I (**A**–**C**), collagen type II (**D**–**F**), and collagen type X (**G**–**I**), in spheroids grown from C28/I2 cells on the RWV after 3, 7, and 14 days. Blue fluorescence: nuclei; orange fluorescence: signals of the respective collagen; arrows: sites of collagen expression.

**Figure 4 biomolecules-14-00025-f004:**
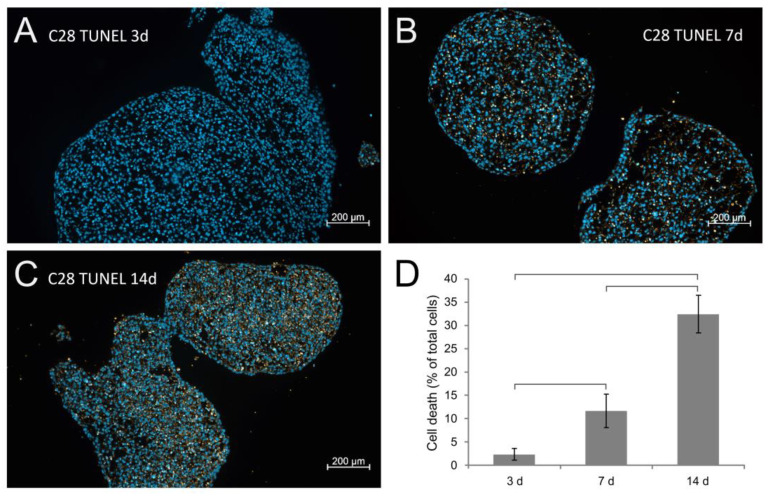
Development of cell death in C28/I2 spheroids over the course of the 14-day cultivation. Sections of harvested spheroids were TUNEL stained, and the amount of orange fluorescence was quantified. Three representative sections are displayed for 3 days (**A**), 7 days (**B**), and 14 days (**C**) and the quantification results are given in (**D**). Cell nuclei were stained with DAPI. Brackets indicate *p* < 0.05.

**Figure 5 biomolecules-14-00025-f005:**
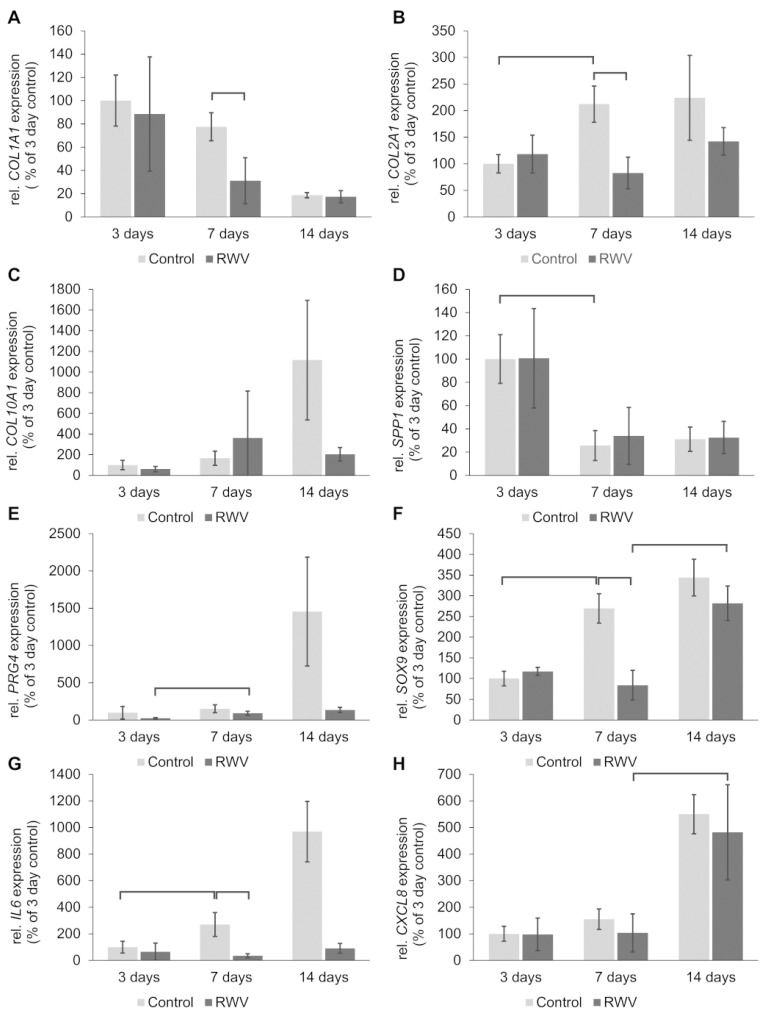
qPCR analyses of chondrocyte-specific and other potetial target genes in spheroids grown from C28/I2 cells on the RWV for 3, 7, and 14 days. Light grey: 1 *g* controls, dark grey: RWV samples; brackets above the bars indicate statistically significant differences with *p* < 0.05. N = 4.

**Figure 6 biomolecules-14-00025-f006:**
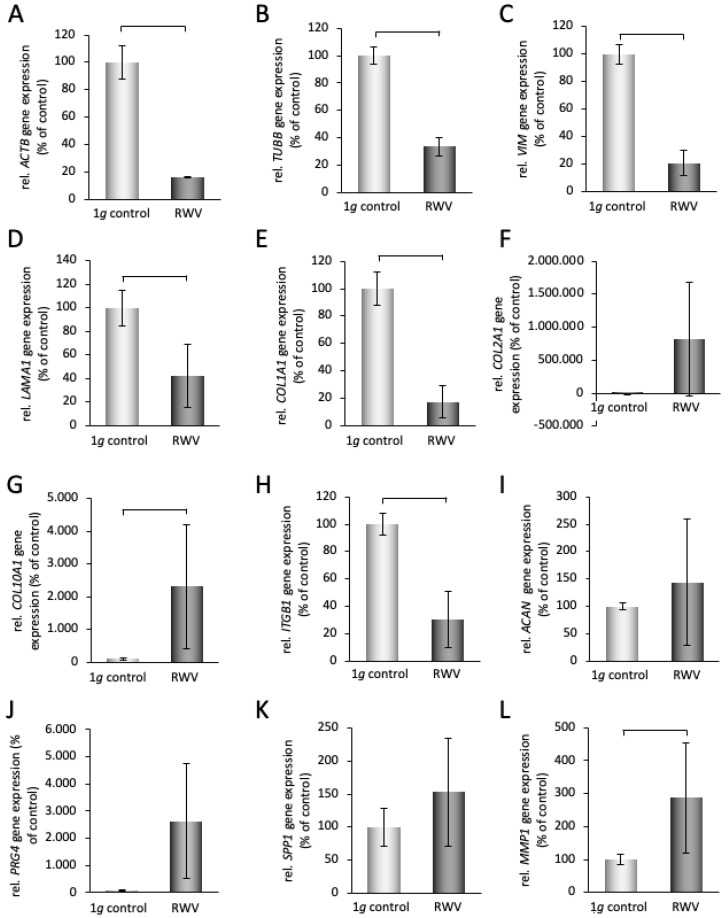
qPCR analyses of chondrocyte-specific and other potetial target genes in spheroids grown from primary human chondroctes on the RWV for 14 days. Light grey: 1 *g* controls, dark grey: RWV samples; brackets above the bars indicate statistically significant differences with *p* < 0.05. N = 5 for RWV and N = 10 for 1 *g* controls.

**Figure 7 biomolecules-14-00025-f007:**
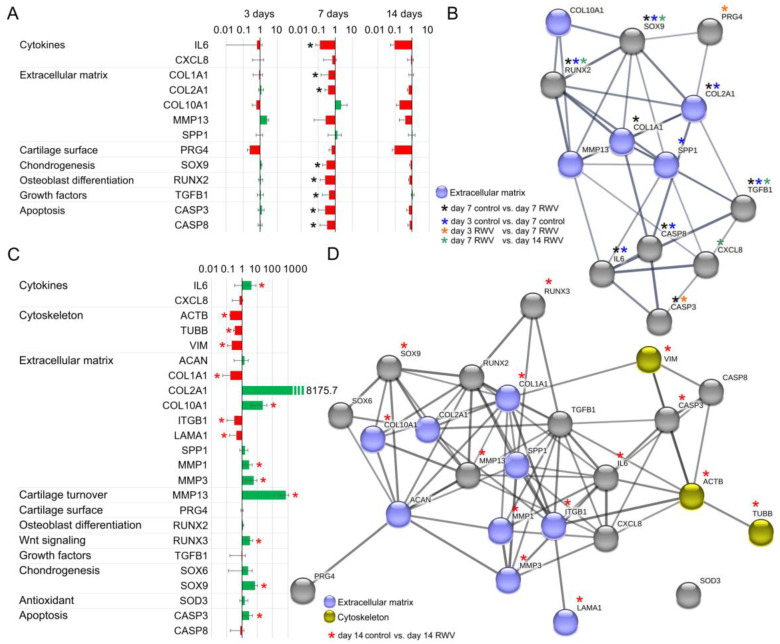
Gene expression overview (**A**,**C**) and EMBL String protein interaction representation (**B**,**D**) of real time qPCR gene expression results for 3D C28/I2 spheroids (**A**,**B**) and primary human chondrocyte spheroids (**C**,**D**). Nominal significant expression differences (*p*-value < 0.05) are marked with asterisks, functionally related genes are color coded.

**Table 1 biomolecules-14-00025-t001:** Primers Used for qPCR Analyses.

Gene	Forward Primer	Reverse Primer
*18S rRNA*	GGAGCCTGCGGCTTAATTT	CAACTAAGAACGGCCATGCA
*ACAN*	TTCTGCTTCCGAGGTGTGTC	CCACCTGAGTGACGATCCAG
*ACTB*	TGCCGACAGGATGCAGAAG	GCCGATCCACACGGAGTACT
*CASP3*	AACTGCTCCTTTTGCTGTGATCT	GCAGCAAACCTCAGGGAAAC
*CASP8*	TGCAAAAGCACGGGAGAAAG	CTCTTCAAAGGTCGTGGTCAAAG
*COL10A1*	GGGCAGAGGAAGCTTCAGAAA	TCTCAGATGGATTCTGCGTGC
*COL1A1*	ACGAAGACATCCCACCAATCAC	CGTTGTCGCAGACGCAGAT
*COL2A1*	GGCAATAGCAGGTTCACGTACA	CGATAACAGTCTTGCCCCACTT
*CXCL8*	TGGCAGCCTTCCTGATTTCT	GGGTGGAAAGGTTTGGAGTATG
*IL6*	CGGGAACGAAAGAGAAGCTCTA	GAGCAGCCCCAGGGAGAA
*ITGB1*	GAAAACAGCGCATATCTGGAAATT	CAGCCAATCAGTGATCCACAA
*LAMA1*	TGACTGACCTGGGTTCAGGA	TGCTAGCACTCCTTGCTTCC
*MMP1*	GTCAGGGGAGATCATCGGG	GAGCATCCCCTCCAATACCTG
*MMP13*	GGAGCCCTGATGTTTCCCAT	GTCTTCATCGCCTGGACCATA
*MMP3*	ACAAAGGATACAACAGGGACCAA	TAGAGTGGGTACATCAAAGCTTCAGT
*PRG4*	CCCCCAAACCACCAGTTGTA	ACGTGTCAGGAGTTGTGACC
*RUNX2*	TGATGACACTGCCACCTCTG	CCAGTTCTGAAGCACCTGCC
*RUNX3*	GTGGGCGAGGGAAGAGTTTC	CCTTGATGGCTCGGTGGTAG
*SOD3*	CTGGAAAGGTGCCCGACTCC	ATGTCTCGGATCCACTCCGC
*SOX6*	GCCACACATTAAGCG	TCCAGCGAGATCCTAAGATTTTG
*SOX9*	AGGAAGTCGGTGAAGAACGG	CGCCTTGAAGATGGCGTTG
*SPP1*	CGAGGTGATAGTGTGGTTTATGGA	CGTCTGTAGCATCAGGGTACTG
*TGFB1*	CACCCGCGTGCTAATGGT	AGAGCAACACGGGTTCAGGTA
*TUBB*	CTGGACCGCATCTCTGTGTACTAC	GACCTGAGCGAACAGAGTCCAT
*VIM*	TTCAGAGAGAGGAAGCCGAAAAC	AGATTCCACTTTGCGTTCAAGGT

All sequences are given in 5′-3′direction.

## Data Availability

Data are contained within the article.

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
