# Peer review of "Structural and Molecular Changes of Human Chondrocytes Exposed to the Rotating Wall Vessel Bioreactor"

_biomolecules, 2023, doi:10.3390/biom14010025_

Round 1

Reviewer 1 Report

Comments and Suggestions for Authors

This paper reported the work of “Structural and Molecular Changes of Human Chondrocytes Exposed to the Rotating Wall Vessel Bioreactor”. This work mainly investigated the process of cartilage formation by primary articular cells and C28/I2 cells at different time stages in microgravity environment using the RWV. The spheroid formation in both cell types starting at day 3. After 14 days constructs from C28/I2 cells and primary articular cells had diameters of up to 5 mm and 3 mm, respectively. The histological morphology, TUNEL-assays and expression of chondrogenic proteins and genes were analyzed by histological staining, immunohistochemical detection of collagen types and quantitative real-time PCR. Overall, the research presented by the authors are useful and interesting. There are a few concerns that should be addressed.

1. The second preamble delves into the discussion of bioprinting methods and bioreactors for cartilage tissue engineering, necessitating a further exploration of their respective advantages and disadvantages. This will subsequently elucidate why the rotating wall vessel (RWV) bioreactor is employed in this study to investigate cartilage tissue formation, thereby enhancing the clarity of the article statement。

2. The study investigated the control groups for normal gravity manipulation, which should be depicted in Figure 1 and compared with the RWV groups.

3. In the genetic tests, intriguing findings were observed: the gene expression profile of newly formed chondrocytes in C28/I2 cells exhibited an inverse pattern compared to that of primary articular chondrocytes. The authors further provided a comprehensive explanation for these results. What would be the result of gene expression in the chondrocytes of the model animals in microgravity?

4. The formatting of the references is inconsistent. For instance, reference 12 lacks an abbreviation for the journal title. Please ensure consistency across all references.

Comments on the Quality of English Language

Good

Author Response

We would like to sincerely thank the reviewers for their valuable comments, which have helped to improve the manuscript substantially. The pointwise changes made are mentioned below.

Reviewer 1

Comments and Suggestions for Authors

This paper reported the work of “Structural and Molecular Changes of Human Chondrocytes Exposed to the Rotating Wall Vessel Bioreactor”. This work mainly investigated the process of cartilage formation by primary articular cells and C28/I2 cells at different time stages in microgravity environment using the RWV. The spheroid formation in both cell types starting at day 3. After 14 days constructs from C28/I2 cells and primary articular cells had diameters of up to 5 mm and 3 mm, respectively. The histological morphology, TUNEL-assays and expression of chondrogenic proteins and genes were analyzed by histological staining, immunohistochemical detection of collagen types and quantitative real-time PCR. Overall, the research presented by the authors are useful and interesting. There are a few concerns that should be addressed.

  1. The second preamble delves into the discussion of bioprinting methods and bioreactors for cartilage tissue engineering, necessitating a further exploration of their respective advantages and disadvantages. This will subsequently elucidate why the rotating wall vessel (RWV) bioreactor is employed in this study to investigate cartilage tissue formation, thereby enhancing the clarity of the article statement.

Answer: Thank you for this important suggestion. We have changed the introduction accordingly and added new references (ref. 11-14; ref. 17,18). Please see page 2, lines 68-79 and lines 83-88.

  1. The study investigated the control groups for normal gravity manipulation, which should be depicted in Figure 1 and compared with the RWV groups.

Answer: Thank you very much. We have included the control groups in Figure 1 and revised the figure legend accordingly.

  1. In the genetic tests, intriguing findings were observed: the gene expression profile of newly formed chondrocytes in C28/I2 cells exhibited an inverse pattern compared to that of primary articular chondrocytes. The authors further provided a comprehensive explanation for these results. What would be the result of gene expression in the chondrocytes of the model animals in microgravity?

Answer: Thank you very much for this suggestion. We have included a new paragraph in the discussion to address this point. New references were added (ref. 69-72). Please see pages 20-21, lines 641-653.

  1. The formatting of the references is inconsistent. For instance, reference 12 lacks an abbreviation for the journal title. Please ensure consistency across all references.

Answer: Thank you for this suggestion. We agree and have checked the references. The abbreviation for the journal title is given. We have used Endnote for the references of this manuscript.

Comments on the Quality of English Language

Good

Answer: Reviewer 2 asked for English editing. Dr. Petra Wise, USC, Los Angeles, USA has edited English language.

Reviewer 2 Report

Comments and Suggestions for Authors

The research article titled, “Structural and Molecular Changes of Human Chondrocytes Exposed to the Rotating Wall Vessel Bioreactor”, explores the various genetic changes that take place when chondrocytes are cultured in a rotating bioreactor. The manuscript provides an elaborate insight into the changes that could be exploited in the future to successfully culture 3D cartilage spheroids, which could eventually be used for various purposes such as drug screening or tissue engineering. The manuscript is well written. The manuscript can be accepted after minor revisions as follows.

1.      The authors could discuss a few lines about the novelty of this work, when compared to other similar works.

2.      Figure 1, scale bars are needed in the microscopic images of cultured cells.

3.      Figure 1 legend, “C, D: After 14 on the RWV…” 14 days?

4.      Figure 1 legend, “3,7 and 14 das, respectively.” Spell check.

5.      Figure 2, scale bars are required.

6.      Page 12, Line 334, “No gene expression changes were measured…”, rephrase the sentence. It seems the authors didn’t measure the gene expression. Better to use alternate terms such as noticed or no significant difference etc.

7.      Page 16, line 408, “Chondrocytes exposed to ….”, Was it the same C28/I2 chondrocytes? The authors could mention the exact cell line name.

Comments on the Quality of English Language

Minor spell checks and grammatical revisions are required.

Author Response

We would like to sincerely thank the reviewers for their valuable comments, which have helped to improve the manuscript substantially. The pointwise changes made are mentioned below.

Reviewer 2

Comments and Suggestions for Authors

The research article titled, “Structural and Molecular Changes of Human Chondrocytes Exposed to the Rotating Wall Vessel Bioreactor”, explores the various genetic changes that take place when chondrocytes are cultured in a rotating bioreactor. The manuscript provides an elaborate insight into the changes that could be exploited in the future to successfully culture 3D cartilage spheroids, which could eventually be used for various purposes such as drug screening or tissue engineering. The manuscript is well written. The manuscript can be accepted after minor revisions as follows.

  1. The authors could discuss a few lines about the novelty of this work, when compared to other similar works.

 Answer: Thank you for this suggestion. We have added this information to the discussion. Please see page 16, lines 391-399.

  1. Figure 1, scale bars are needed in the microscopic images of cultured cells.

 Answer: Yes, we agree and have included the scale bars in the Phase contrast images of Figure 1.

  1. Figure 1 legend, “C, D: After 14 on the RWV…” 14 days?

Answer: Thank you. The error in the Figure legend 1 was corrected.

  1. Figure 1 legend, “3,7 and 14 das, respectively.” Spell check.

Answer: Thank you. The error on the Figure legend 1 was corrected.

  1. Figure 2, scale bars are required.

Answer: Yes, we agree and have included the scale bars in the images of Figure 2.

  1. Page 12, Line 334, “No gene expression changes were measured…”, rephrase the sentence. It seems the authors didn’t measure the gene expression. Better to use alternate terms such as noticed or no significant difference etc.

Answer: Thank you. The sentence was rephrased. Please see page 12, now lines 350-351.

  1. Page 16, line 408, “Chondrocytes exposed to ….”, Was it the same C28/I2 chondrocytes? The authors could mention the exact cell line name.

Answer: Thank you. We have given the exact cell line name in the text (Please see page 16, line 432).

Comments on the Quality of English Language

Minor spell checks and grammatical revisions are required.

Dr. Petra Wise, Children’s Hospital, USC, Los Angeles, USA edited English language of the revised manuscript.

Round 2

Reviewer 1 Report

Comments and Suggestions for Authors

The paper can be accepted after the revision.